# Distribution and Driving Force of Water Use Efficiency under Vegetation Restoration on the Loess Plateau

**Ruixue Ma** [1,2], **Dacheng Wang** [2,*], **Ximin Cui** [1], **Xiaojing Yao** [2], **Shenshen Li** [2], **Hongsen Wang** [1] **and Bingxuan Liu** [3]

[1] College of Geoscience and Surveying Engineering, China University of Mining & Technology, Beijing 100083, China
[2] Aerospace Information Research Institute, Chinese Academy of Sciences, Beijing 100094, China
[3] Scientific and Technological Innovation and Development Center of the Chinese Academy of Sciences, Beijing 100045, China
[*] Correspondence: wangdc@aircas.ac.cn

**Abstract:** The Grain for Green Project (GGP) has considerably improved the vegetation cover of the Loess Plateau, as well as changed the carbon and water coupling process of local vegetation to a certain extent. Water use efficiency (WUE) is a crucial measure for evaluating ecosystem responses to global climate change and is a key indicator of the carbon–water coupling between terrestrial ecosystems and the environment. A comprehensive understanding of the impact of vegetation reconstruction on WUE on the Loess Plateau is of great significance to the vegetation growth and contribution to sustainable of the Loess Plateau. In recent years, scholars have gained a more comprehensive understanding of the distribution and drivers of WUE on the Loess Plateau. However, through the study of carbon and water coupling in the Loess Plateau, it is found that the effects of different vegetation restoration levels on WUE are still to be studied in depth in terms of spatial and temporal heterogeneity and long timeseries. In this paper, we analyzed the trends of Normalized Difference vegetation cover (NDVI) and WUE from 2001 to 2010 and 2011 to 2020, respectively, to research at the WUE of the vegetation in this area in relation to vegetation restoration. It was found that the Loess Plateau's vegetation WUE rose from 2001 to 2020 at a rate of 0.023 g C kg$^{-1}$ H$_2$O per year, and that the increase from 2011 to 2020 was more significant than the growth from 2000 to 2010. The Loess Plateau's area with a growing trend in vegetation water use rate increased from 77.12% in 2001–2010 to 88.63% in 2011–2020, with the majority of the increased area occurring in the northeastern Inner Mongolia region. After 20 years of the reforestation project, the area where NDVI and WUE increased simultaneously accounted for 71.54% of the Loess Plateau, the area where NDVI increased but WUE decreased accounted for 10.95% of the Loess Plateau, and the area where NDVI increased but WUE decreased accounted for 7.15% of the Loess Plateau. The correlation between temperature precipitation and WUE was not significant for the whole Loess Plateau, further indicating that the increase in vegetation cover was the main reason for the increase in vegetation water efficiency. Therefore, the effect of vegetation cover on WUE should be fully considered when vegetation restoration is carried out on the Loess Plateau.

**Keywords:** the Loess Plateau; water use efficiency (WUE); Normalized Difference vegetation cover (NDVI); Grain for Green Project (GGP); remote sensing; precipitation; temperature

## 1. Introduction

Ecosystem water use efficiency (WUE), which is typically defined as the quantity of dry matter fixed per unit mass of water absorbed by terrestrial ecosystems, is a measure of ecosystem carbon–water interactions [1], calculated as the ratio of gross primary production (GPP) to evapotranspiration (ET) [2]. WUE not only reflects the vegetation carbon and water cycle mechanism and its coupling relationship, but also a thorough physiological and

ecological index to assess if plant growth is suitable [3]. The characteristics of WUE depend mainly on the strength of the coupling of the GPP and ET components, which are important variables linking the carbon cycle and hydrological cycle of vegetated ecosystems [4]. The research on the long time series of WUE can help to explore the impact of climate change behind the ecosystem changes at a macro level and reveal the mechanism of global changes on the interaction relationship between water and carbon cycles.

The Yellow River Basin, a typical arid and semiarid region, contains the middle and upper sections of the Loess Plateau, the world's largest loess accumulation area [5]. Due to the long-term unreasonable use of land by human beings, the original carbon and water balance has been destroyed, which, together with the loose soil of the Loess Plateau, makes the ecosystem in a long-term unstable state, with strong storm washing and severe soil erosion, making it a typical ecologically vulnerable region in China [6]. Since the beginning of the "Grain for Green" project (GGP) in 1999, many national policies have been introduced to restrict land degradation on the Loess Plateau, and, since 2000, the Loess Plateau has been in the ecological restoration stage. The ecological environment of the Loess Plateau has been greatly improved by the large-scale implementation of the GGP to forests and grasses and building green water and green mountains [7]. As a typical area of aridity and semi-aridity, the Loess Plateau has precipitation as the main source of water, often due to insufficient precipitation resulting in low groundwater reserves in the area, and the growth of large vegetation needs to absorb and use a large amount of water, thus leading to excessive water consumption, carbon and water imbalance, and some vegetation degradation [8]. Therefore, it is important to investigate the effects of vegetation cover and precipitation temperature on vegetation WUE to promote the future-oriented ecological restoration management system and modernization of governance capacity in Loess Plateau.

In recent years, many scholars have conducted numerous studies on plant WUE using remote sensing methods. For example, Huang et al. [9] found that WUE increases mainly at high latitudes and decreases at middle and low latitudes under the context of current climate change. Tang et al. [10] calculated WUE using MODIS data and showed that the latitudinal distribution of WUE increases from subtropical to middle and high latitudes and then decreases with increasing latitude. Scartazza et al. [11] used eddy correlation techniques and stable isotopes to measure the WUE of forest ecosystems at different scales in the Mediterranean region. Zhao et al. [12] analyzed the mechanism of WUE on precipitation temperature and drought in the Hai River basin in China using MODIS data. Cao et al. [13] studied the mechanism of soil water content underlying the efficiency of vegetation water utilization in China. Since the national implementation of ecological restoration projects such as the return of cultivated land to forest and grass in 1999, large-scale ecological restoration has significantly changed the regional ecohydrological processes.

Research has been conducted on vegetation types and climate change in the Loess Plateau. Zhang et al. [14] studied the Loess Plateau's vegetation growth over a lengthy timeseries from 1982 to 2014 using AVHRR and MODIS NDVI data. In order to accurately assess the ecological effects of reforestation, Xin et al. [15] investigated the spatial and temporal history of vegetation cover on the Loess Plateau and its reaction to meteorological conditions using GIMMS NDVI data, and they discovered that the study area's overall vegetation cover showed an increasing tendency and a high association with climate. Liu et al. used 54 meteorological stations on the Loess Plateau from 1957 to 2012 to study the climate aridity of the Loess Plateau [16]. However, these studies focus on the Loess Plateau as a whole within a certain time period, or only compare before and after GGP, and rarely compare the changes of WUE in the Loess Plateau during different periods of fallowing. In fact, due to the different rates of vegetation restoration, different water–carbon coupling mechanisms are presented at different time periods, and balancing carbon uptake and water consumption at a regional scale is a challenge.

Previous studies struggled to investigate the WUE of the entire Loess Plateau and its drivers but cannot be conducted for the complex spatial heterogeneity. Therefore, in this paper, we used PML remote sensing GPP and ET products, combined with normalized

vegetation index (NDVI) and meteorological data, to study the effect of different vegetation cover degrees on vegetation WUE for the Loess Plateau region, facing complex spatial heterogeneity and a long timeseries, and we also investigated the main driving forces affecting the WUE to provide a scientific basis for the sustainable development of vegetation on the Loess Plateau.

## 2. Model Approach

### 2.1. Introduction to the Study Area

The Loess Plateau is located at 103–114°E, 34–40°N, from Taihang Mountains in the east to the Sun and Moon Mountains in the west, reaching the Great Wall in the north and the Qinling Mountains in the south, mainly including parts of Shanxi, Shaanxi, Gansu, Qinghai, Ningxia, and Henan provinces [17]. The total area of the region is 63.5 × 10⁴ km², taking up 6.44% of China's total land area (Figure 1). An average annual temperature of 3.6–14.3 °C characterizes its temperate continental monsoon climate. Winter and spring are influenced by polar air masses that are dry and chilly, making it frigid, dry, and sandy; summer and autumn are hot and heavy, with an average annual rainfall of 466 mm that is primarily concentrated in June to September, accounting for almost 60% of the annual precipitation. Winter precipitation generally accounts for only about 5% [18]. Long-term excessive deforestation and grazing have made the Loess Plateau region the worst example of soil erosion and the weakest ecological condition in China and the world [19]. The vegetation in the study area from south to north shows a general trend of overgrowth from forest to grassland. The central region consists primarily of a semiarid grassland zone. The topography progressively changes to a desert in the northwest, with desert grassland predominating [20]. The fragile ecological environment combined with long-term unreasonable land reclamation and utilization by humans has exacerbated the ecological problems of the Loess Plateau. Since the implementation of the reforestation project, the planting of large areas of vegetation has led to a trend of overall improvement and local degradation of regional vegetation [21].

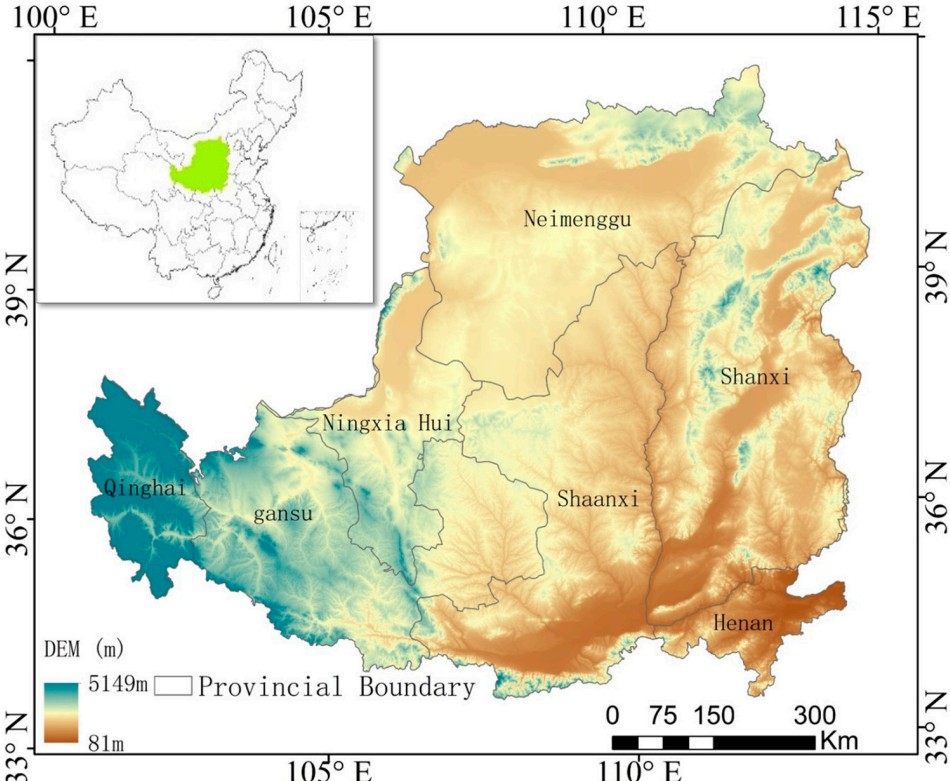

**Figure 1.** The location map of the Loess Plateau.

### 2.2. Data Sources

This paper's GPP and ET data were acquired utilizing the GEE platform PML_V2v0.1.7 data product (https://code.earthengine.google.com/, accessed on 18 July 2021). Firstly, the PML_V2 image 1840 scenes passing over the Loess Plateau from 2001 to 2020 were acquired in the GEE platform using the filtering functions of time and space; the images were processed by cropping and removing the outliers, and the GPP, ET effects were fused for each year using the cumulative method. The PML-V2 model is based on the stomatal conductivity theory coupled with the total primary productivity process [22]. The parameters were determined and parameterized on 95 global eddy-related flux stations according to different vegetation types, and then extended globally based on land use data. To ensure the accuracy of the model simulation results, this thesis was validated using data from the FLUXNET2015 site (https://fluxnet.org/data/fluxnet2015-dataset/, accessed on 26 June 2022). Water–carbon coupling showed high accuracy in the study local area. NDVI data were obtained from the MOD13A2V6 product (https://code.earthengine.google.com/, accessed on 26 June 2022). DEM data were obtained from the Shuttle Radar Topography Mission (SRTM). The SRTM elevation data were obtained from the Interferometric Synthetic Aperture Radar (InSAR) sensor, (http://srtm.csi.cgiar.org/, accessed on 20 June 2022). The data used in this study are shown in Table 1.

**Table 1.** Research data sources.

| Product Model | Surface Parameters | Time Resolution | Spatial Resolution |
|---|---|---|---|
| PMLV2v0.1.7 | GPP | 8 d | 500 m |
| PMLV2v0.1.7 | ET | 8 d | 500 m |
| MOD13A2 | NDVI | 16 d | 500 m |
| SRTM V4.1 | DEM | - | 90 m |

The National Meteorological Science Data Center provided the meteorological information utilized in this paper (http://data.cma.cn/, accessed on 26 June 2022). Fifty-two meteorological stations, including precipitation and average temperature data, are involved within the Loess Plateau in seven provinces: Shanxi, Qinghai, Ningxia, Inner Mongolia, Henan, Gansu, and Shaanxi.

### 2.3. Research Methods

Currently, there are different definitions and understandings of vegetation WUE in academia, and the results are expressed in different ways [23]. In this paper, the ratio of GPP to ET was taken as the most classical way to calculate WUE. The GPP and ET product data have been validated by flux tower site data in several regions of the world, and their accuracy has been confirmed in several studies [24]. The WUE is calculated as follows:

$$WUE = \frac{GPP}{ET},\tag{1}$$

where WUE is the water use efficiency (g C·kg$^{-1}$ H$_2$O), GPP is the total primary productivity of terrestrial ecosystems (g C·m$^{-2}$), and ET is the evaporation of ecosystems (kg H$_2$O·m$^{-2}$).

### 2.4. Analysis Method

#### 2.4.1. Trend Analysis

Trend analysis is a technique for analyzing patterns in timeseries statistically [25]. Theil-Sen median (SEN) was utilized in this study to further evaluate the trend in changing environmental variables in the studied area. Since the data need not follow a specific distribution and are resistant to errors, SEN trend analysis produces more accurate calculation

results; it is now widely used in hydrology and climate timeseries research [26]. Its formula is as follows:

$$\beta = Median\left(\frac{X_j - X_i}{j - i}\right) \forall j > i,$$ (2)

where *Median ()* represents the median value, $X_j$ and $X_i$ are elements of the time series of the trend to be analyzed, in chronological order. If $\beta$ is greater than zero, it represents an upward trend in the timeseries; otherwise, it represents a downward trend.

### 2.4.2. Coefficient of Variation

The coefficient of variation (CV) is a statistic that quantifies the degree of variation in the observed sequence value, which can reflect the degree of variation in the geographic data in the timeseries and allow an assessment of the stability of the data timeseries. A larger value indicates worse stability [27]. The calculation formula is as follows:

$$\mu = \frac{1}{N}\sum_{i=1}^{N} x_i,$$ (3)

$$\sigma = \sqrt{\frac{1}{N}\sum_{i=1}^{N}(x_i - \mu)^2},$$ (4)

$$CV = \frac{\sigma}{\mu},$$ (5)

where $\mu$ is the mean value, and $\sigma$ is the standard deviation. $N$ represents the total number of data, $Xi$ represents the $i$-th element. A smaller $CV$ indicates better stability.

### 2.4.3. Correlation Analysis

In this research, the correlation between each element was examined using the Pearson coefficient, which may be determined as follows:

$$R = \frac{\sum(x - \bar{x})(y - \bar{y})}{\sqrt{\sum(x - \bar{x})^2 \sum(y - \bar{y})^2}},$$ (6)

where $R$ is the Pearson coefficient, $x$ is the influence factor, $y$ is the dependent variable, and $\bar{x}, \bar{y}$ indicate the average of $x, y$, respectively. The size of the $R$ value reflects the correlation of the relationship between the factors, whereby positive correlations are denoted by $R > 0$ and adverse correlations by $R < 0$.

## 3. Results

### 3.1. Data Verification

In this paper, the 8 day and month-by-month PML remote sensing products (GPPP$_{PML}$, ET$_{PML}$) and flux site observation data (GPP$_{OBS}$, ET$_{OBS}$) were fitted, and the results showed good agreement for both datasets. For the 8 day scale data, the $R^2$ of GPP was 0.94 and the RMSE was 0.74 g C m$^{-2}$/8 days, while the $R^2$ of ET data was 0.82 and the RMSE was 0.29 mm/8 days (Figure 2a,c); for the monthly scale data, the $R^2$ of GPP was 0.92 and the RMSE was 28.47 g C m$^{-2}$, while the $R^2$ of ET data was 0.74 and the RMSE was 8.56 mm (Figure 2b,d). The results show that the fitting accuracy between the two datasets was high; hence, it was feasible to use PML data to study the carbon–water coupling process in the Loess Plateau.

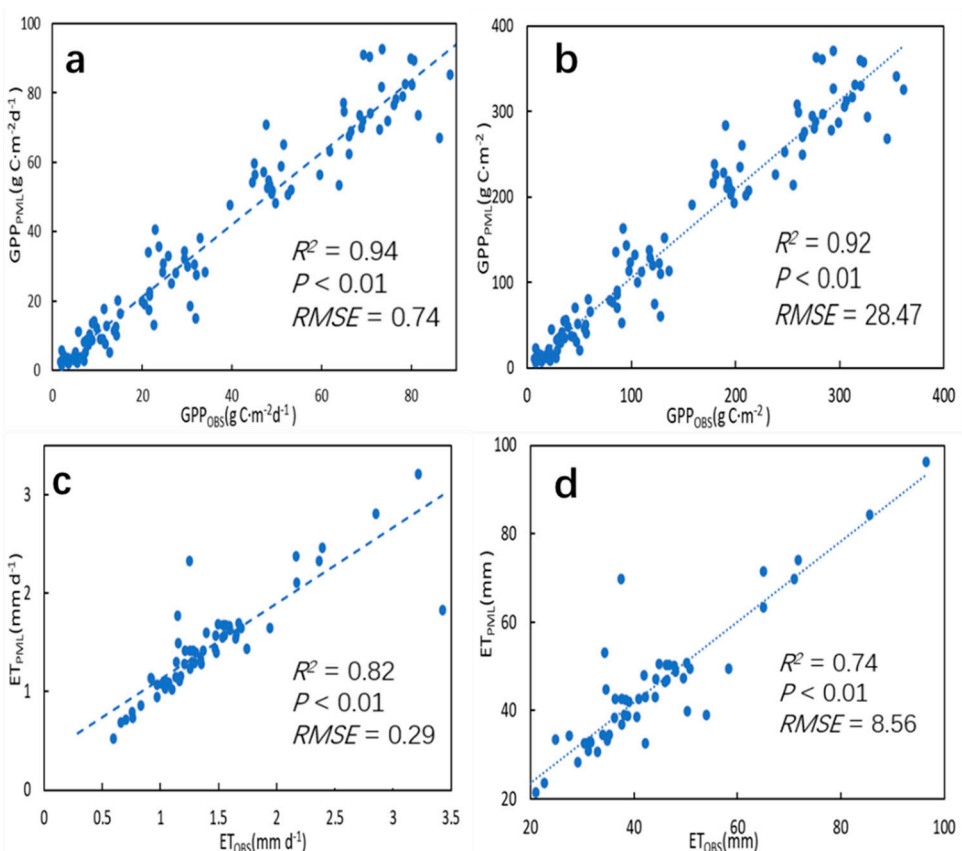

**Figure 2.** Validation diagrams of GPP and ET remote sensing products of PML data at 8 day scale (**a**,**c**) and monthly scale (**b**,**d**).

### 3.2. NDVI Annual Variation on the Loess Plateau

The mean NDVI value for 2011–2020 on the Loess Plateau was 0.58, slightly higher than 0.51 for 2001–2010, and the CV value for 2011–2020 was lower than that for 2001–2010, indicating a more uniform temporal distribution of NDVI for 2011–2020 (Table 2). The average annual increase rate of NDVI for 2011–2020 was 0.008, compared with 0.01 from 2001 to 2010, indicating a slight decrease in the increase rate.

**Table 2.** Statistical table of the value and growth rate of NDVI in Loess Plateau.

|  | Period | 2001–2010 | 2011–2020 |
|---|---|---|---|
| | Min | 0.007 | 0.010 |
| | Max | 0.940 | 0.946 |
| value | Mean | 0.507 | 0.584 |
| | STD | 0.203 | 0.198 |
| | CV | 0.062 | 0.052 |
| | Min | −0.090 | −0.097 |
| | Max | 0.090 | 0.099 |
| Growth rate | Mean | 0.007 | 0.006 |
| | STD | 0.009 | 0.008 |
| | CV | 0.066 | 0.058 |

From 2001 to 2010, 81% of the areas on the Loess Plateau showed an increasing trend in NDVI, while only some areas in northeastern Inner Mongolia and a few areas in southwestern Gansu showed a slight decrease in NDVI. Among them, 18.3% of the areas showed a significant increase, mainly in Yan'an, Yulin, and Linfen, which also correspond to the areas with the most obvious vegetation restoration in the fallow forest project (Figure 3a).

This indicates that the initial revegetation of the Loess Plateau has achieved certain results. Compared with 2001–2010, the area of NDVI increase in 2011–2020 moved from southeast to northwest, and the area share increased from 81% to 83%, mainly in Hohhot, Inner Mongolia, and Pingliang, Gansu (Figure 3b). However, the vegetation cover in the Shanxi plateau, as well as the Guanzhong plain area, showed a decreasing trend from 2011 to 2020.

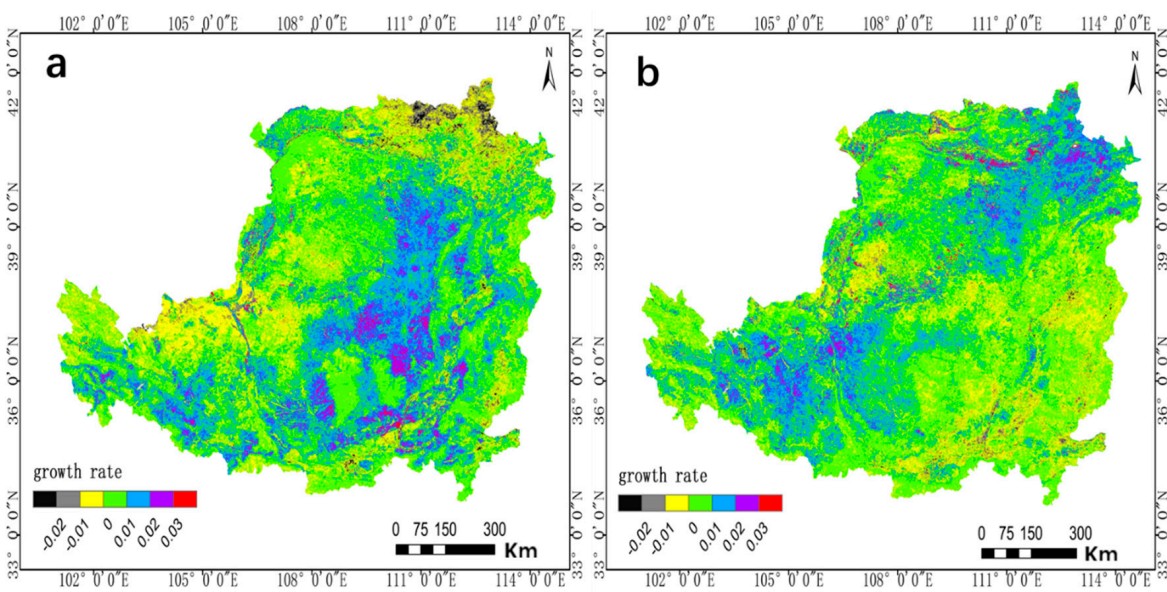

**Figure 3.** (**a**) Variation trend of NDVI in the Loess Plateau from 2001 to 2010. (**b**) Variation trend of NDVI in the Loess Plateau from 2011 to 2020.

### 3.3. Variation of WUE on the Loess Plateau

The mean WUE of the Loess Plateau from 2001 to 2010 was 1.12 g C·kg$^{-1}$ H$_2$O, increasing at an average rate of 0.025 g C·kg$^{-1}$ H$_2$O per year. The 2011–2020 WUE averaged 1.35 g C·kg$^{-1}$ H$_2$O and increased at a rate of 0.032 g C·kg$^{-1}$ H$_2$O per year (Table 3). The 2011–2020 standard deviation and CV were smaller than the values from 2001–2011, indicating a stronger stability in the last decade.

**Table 3.** Statistical table of WUE value and growth rate in Loess Plateau.

|  | Period | 2001–2010 | 2011–2020 |
|---|---|---|---|
| value | Min | 0.000 | 0.000 |
|  | Max | 29.106 | 29.486 |
|  | Mean | 1.120 | 1.350 |
|  | STD | 0.619 | 0.666 |
|  | CV | 0.081 | 0.082 |
| Growth rate | Min | −0.291 | −0.291 |
|  | Max | 0.247 | 0.423 |
|  | Mean | 0.015 | 0.032 |
|  | STD | 0.025 | 0.035 |
|  | CV | 0.085 | 0.074 |

The area of the Loess Plateau with an increasing trend of WUE from 2001 to 2010 accounted for 77.12%, mainly distributed near the Luliang Mountain Range at the junction of Shaanxi and Shanxi provinces, along with some areas in Qinghai and Gansu (Figure 4a). Compared with 2001–2010, the area with an increasing trend of WUE in 2011–2020 increased by 11.51% and the change was more significant (Figure 4b). Similar to the changes in NDVI, especially in some regions of Inner Mongolia, the WUE trend changed from decreasing to increasing in 2011–2020, indicating that both NDVI and WUE improved more significantly in this region.

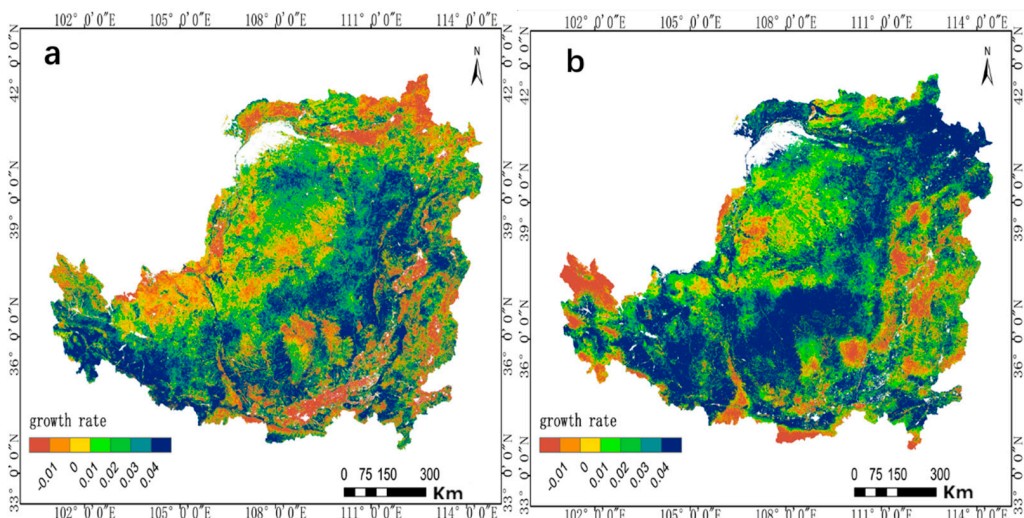

**Figure 4.** (**a**) Trend of WUE of Loess Plateau 2001–2010. (**b**) Trend of WUE of Loess Plateau 2011–2020.

*3.4. The Relationship and Correlation between NDVI and WUE*

In this paper, the changes of NDVI and WUE from 2001 to 2010 and 2011 to 2020 were superimposed. It was found that the area where NDVI and WUE increased at the same time accounted for the largest proportion, and the area of the Loess Plateau from 2001 to 2010 accounted for the largest proportion, increasing from 6.845% to 7.464% in 2011–2020 (Table 4). This shows that the restoration of vegetation on the Loess Plateau has played a certain role in promoting vegetation WUE. The area where NDVI increased and WUE decreased was reduced from 13.08% of the Loess Plateau to 8.81% from 2001 to 2010, mainly reflected in Linhe, Inner Mongolia, and Yulin, Shaanxi (Figure 5a). The area where NDVI decreased but WUE increased rose by 2.98% in the two time periods, mainly reflected in some areas such as Yan'an and Weinan in Shaanxi Province. The area where NDVI and WUE decreased at the same time accounted for 10.12% of the Loess Plateau from 2001 to 2010, but only 5.28% of the total area from 2011 to 2020 (Figure 5b), indicating that both NDVI and WUE of the Loess Plateau have been reduced to a certain extent.

**Table 4.** The changes of NDVI and WUE.

| Period | NDVI | WUE | Area Change Ratio (%) |
|:---:|:---:|:---:|:---:|
| | − | − | 10.12 |
| | − | + | 8.34 |
| 2001–2010 | + | − | 13.08 |
| | + | + | 64.45 |
| | − | − | 5.28 |
| | − | + | 11.28 |
| 2011–2020 | + | − | 8.81 |
| | + | + | 74.63 |

Note: + means increase, − means decrease.

To learn more about the Loess Plateau's NDVI's process for influencing WUE, the correlation coefficients of NDVI with WUE, GPP, and ET in 2001–2010 and 2011–2020 were calculated. It can be seen from the results (Figure 6) that the correlation of NDVI with GPP was higher than with ET, and NDVI mainly led to a change in the WUE by affecting GPP. Different correlations were observed in different regions. The correlation between NDVI and WUE in the same change region was also higher than others. The sensitivity coefficient of NDVI to WUE was 0.59 (Figure 6c), the sensitivity coefficient of NDVI to GPP was 0.68 (Figure 6f), and the sensitivity coefficient of NDVI to ET was 0.32 (Figure 6i). NDVI was positively correlated with WUE, GPP, and ET as a whole; however, in 2011–2020, in the area

where NDVI increased and WUE decreased, NDVI and ET showed a negative correlation with a correlation factor of −0.02.

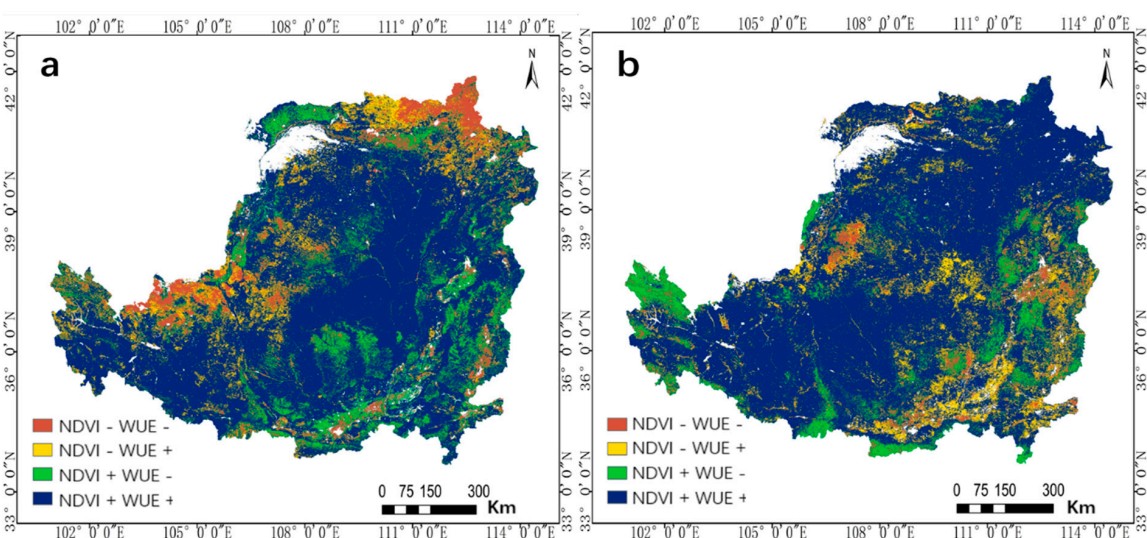

**Figure 5.** (**a**) Distribution map of NDVI and WUE area change from 2001 to 2010. (**b**) Spatial distribution map of NDVI and WUE from 2011 to 2020.

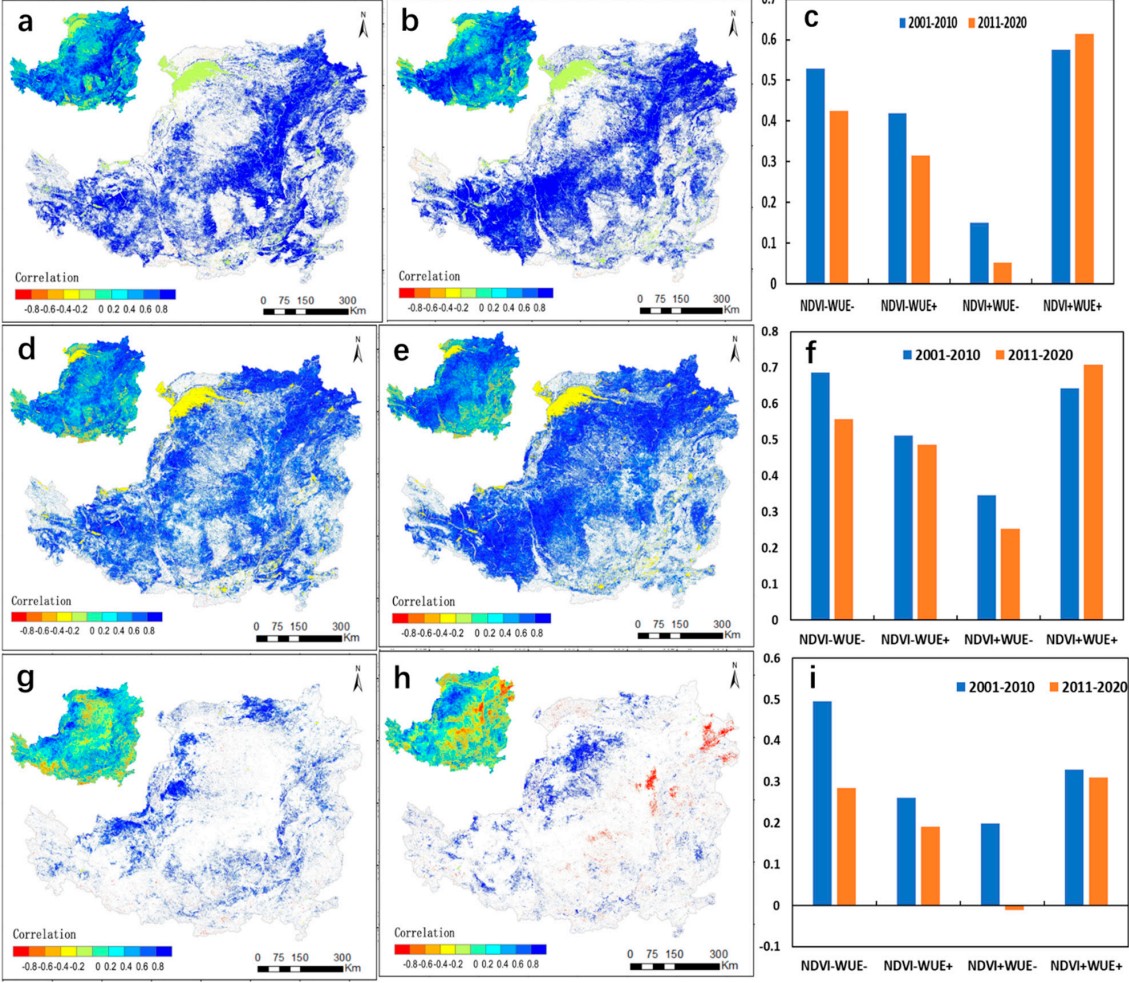

**Figure 6.** Spatial distribution and statistical images of correlation between NDVI and WUE (**a–c**), NDVI and GPP (**d–f**), and NDVI and ET (**g–i**). (**a**,**d**,**g**) are correlations for 2001–2010, (**b**,**e**,**f**) are

correlations for 2011–2020; the upper right corner is the image without excluding significant *p* > 0.05 elements (**c,f,i**). Correlation statistics of NDVI with GPP, ET, and WUE.

### 3.5. WUE Driving Factor Determination

As shown in Figure 7, the precipitation and temperature patterns in the study area changed greatly in the 10 years before and after 2010. After 2010, both precipitation and air temperature on the Loess Plateau increased significantly [28] (Figure 7b,e). The precipitation change rate increased from 1.29 mm/a to 3.89 mm/a, and the temperature increased from 0.03 °C/a to 0.13 °C/a. From 2011 to 2020, the trend of precipitation in the increased WUE changed from decreasing to increasing. The average change of precipitation in this region from 2000 to 2010 was −0.09 mm/a, whereas, from 2011 to 2020, it was 0.27 mm/a (Figure 7f). Similar to the change in precipitation, the air temperature in the area where WUE increased on the Loess Plateau showed a significant upward trend, from −0.01 °C/a to 0.38 °C/a. From 2001 to 2020, the air temperature in the reduced WUE area showed a decreasing trend.

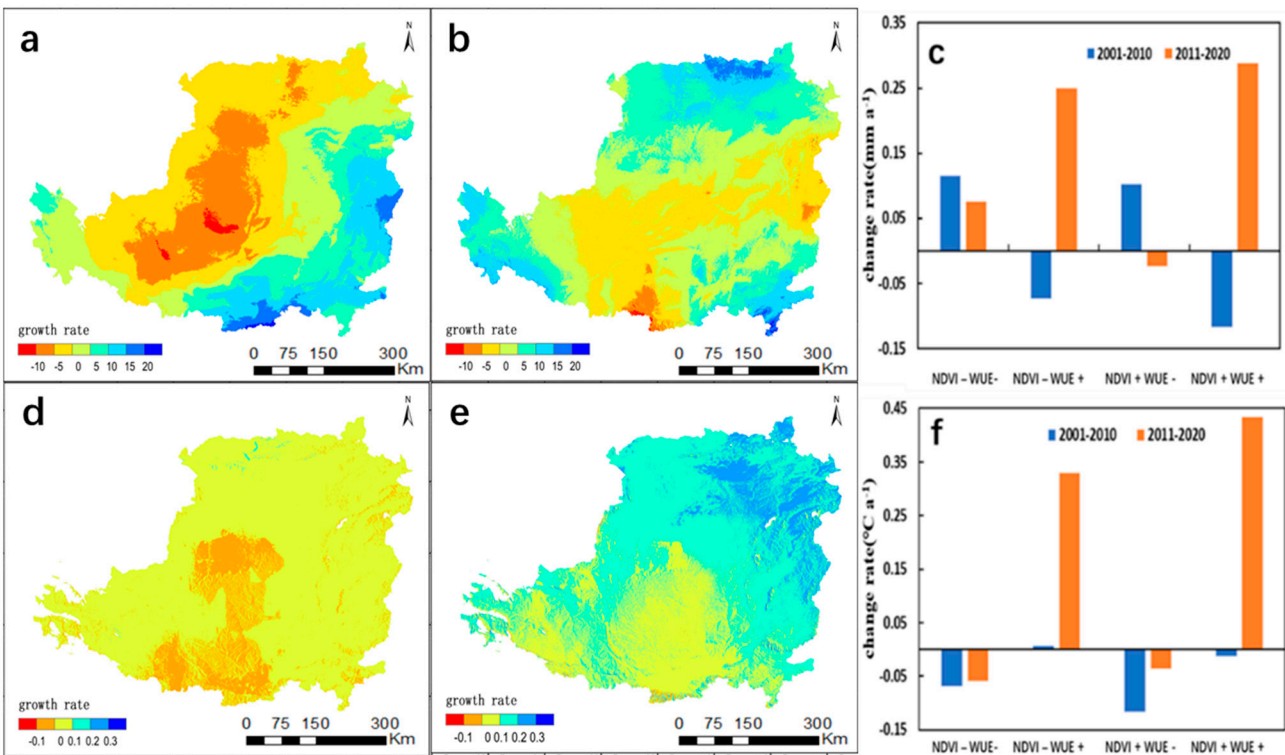

**Figure 7.** (**a**) Variation trend of precipitation on the Loess Plateau from 2001 to 2010. (**b**) Variation of precipitation on the Loess Plateau from 2011 to 2020. (**c**) Statistical graph of precipitation on the Loess Plateau. (**d**) Variation of temperature on the Loess Plateau from 2001 to 2010. (**e**) Trend map of temperature changes in the plateau from 2011 to 2020. (**f**) Statistical map of temperature changes in the Loess Plateau.

In order to test the dependence of WUE on potential driving factors, the correlation distributions between WUE and temperature and precipitation on the Loess Plateau from 2001 to 2010 and 2011 to 2020 were drawn. According to correlation distribution map at the 0.05 significance level, the regions where the precipitation passed the significance test from 2001 to 2010 accounted for 11.42% of the Loess Plateau, and nearly all of them displayed a negative association. (Figure 8a,b). Eastern Inner Mongolia and northern Shaanxi were the areas with the highest connection. Only 4.63% of the areas with temperature passed the significance test, showing an insignificant negative correlation as a whole. From 2011 to

2020, 12.65% of the precipitation areas passed the significance test, and most of the areas that passed the test were positively correlated; the area with the strongest correlation was in the central area where WUE showed an increasing trend, with 17.98% of the areas passing the significant correlation. The correlation was generally positive, mainly distributed in Inner Mongolia in the northeast and parts of Gansu in the southwest.

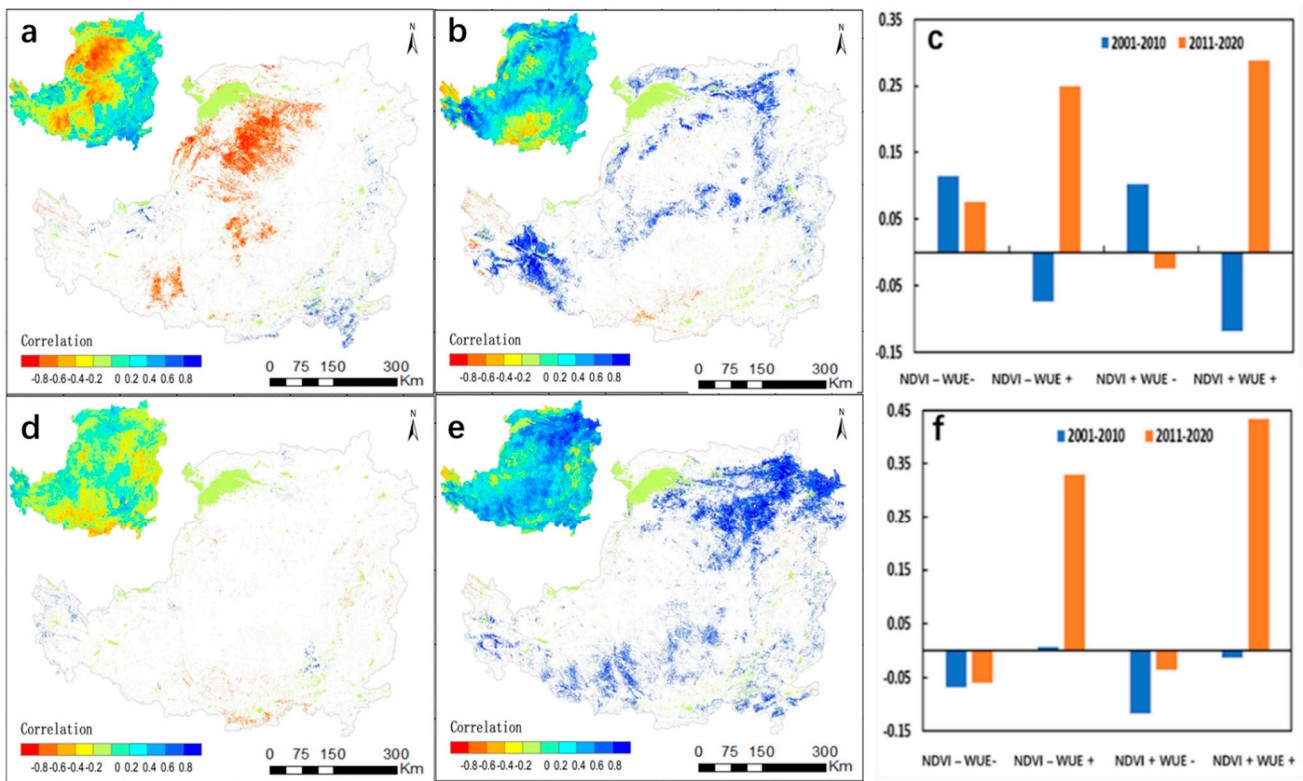

**Figure 8.** Spatial distribution and statistical images of correlation between precipitation and WUE (**a**–**c**), WUE, and temperature (**d**–**f**). (**a**,**d**) are correlations for 2001–2010, (**b**,**e**) are correlations for 2011–2020; The upper right corner is the image without excluding significant *p* > 0.05 elements. (**c**,**f**) are statistical maps of the correlation between WUE and meteorological factors.

## 4. Discussion

The carbon–water coupling process of vegetation is a critical part of the global carbon–water cycle and is affected by both meteorological factors and human activities. Therefore, there are many factors that drive WUE, such as precipitation and temperature, vegetation, soil water content, carbon dioxide concentration, and land-use transformation. Therefore, this study discussed the effects on vegetation growth and carbon–water coupling on the basis of different vegetation coverage and precipitation temperature.

### 4.1. Areas where NDVI and WUE Increased at the Same Time

The statistical findings show that, among the four categories, the area where NDVI and WUE rose simultaneously increased from 68.45% of the entire area of the Loess Plateau in 2001–2010 to 74.64% in 2011–2020. This clearly demonstrates how the Loess Plateau's program of converting cropland to forests has had an impact on vegetation restoration and led to an improvement in vegetation WUE.

With regard to GPP and ET, NDVI displayed a favorable connection. Because of the increased vegetation cover, both GPP and ET were improved to a certain extent, which eventually led to a significant increase in WUE [29]. The precipitation temperature in this region had a relatively obvious increase from 2011 to 2020 (Figure 7), and the temperature and precipitation in this region had a relatively obvious positive correlation. Therefore,

under the significant improvement of various driving factors in recent years, the WUE also increased significantly.

### 4.2. Areas where NDVI Decreased but WUE Increased

The research found that, from 2001 to 2010, 8.34% of the area of the Loess Plateau had reduced vegetation coverage but increased WUE, mainly distributed in Baotou, Hohhot, Lanzhou, and Wuzhong in Inner Mongolia, and the vegetation type was mainly grass [30]. The area of this region rose to 11.28% between 2011 and 2020, gradually encroaching on the Loess Plateau's southeast.

This is due to the high latitude, low temperature, and sufficient sunshine in this area, which is suitable for grassland growth [16,31]. Compared with other vegetation types, grassland has less evapotranspiration, but sufficient precipitation renders the vegetation unable to absorb it [32]. Accordingly, the ET in this area decreases, while the GPP maintained an increasing trend.

### 4.3. Areas where NDVI Increased but WUE Decreased

The results show that, from 2001 to 2010 and 2011 to 2020, the vegetation coverage in the Loess Plateau increased by 13.08% and 8.8%, but the vegetation WUE decreased. The distribution areas were mainly located in Shanxi and Shaanxi, among which the most obvious impact was seen in meadows and woodlands [33].

Due to the relatively high average temperature in the eastern part of the Loess Plateau, with the increase in temperature and light, vegetation transpiration has increased, and soil water evaporation has intensified. Constrained by water, precipitation has become the main limiting factor [34]. However, large-scale vegetation growth requires more water, and precipitation cannot replenish soil water loss in time, resulting in a decline in regional vegetation WUE [35]. From 2001 to 2010, when the policy of returning farmland to forest was implemented, large-scale artificial afforestation led to a significant increase in evapotranspiration at a rate of 6.33 mm per year, but precipitation in the region decreased slightly at a rate of 0.028 mm per year, thus becoming a major factor for the decline in regional vegetation WUE. In 2010–2020, compared with 2001–2010, the area of increased WUE decreased by 4.28%, and the overall distribution moved from the south to the east. This is because some grasslands were transformed into cultivated land after 2010, and artificial irrigation also provided a certain amount of water for the surrounding vegetation, which alleviated this phenomenon to a certain extent.

### 4.4. Areas where NDVI and WUE Decreased Simultaneously

Studies showed that the vegetation coverage and vegetation WUE of the Loess Plateau decreased by 10.12% and 5.28% from 2000 to 2010 and from 2010 to 2020, respectively. This area gradually increased by 200 mm from Baotou in Inner Mongolia, Lanzhou in Gansu, and Wuzhong in Ningxia to Shanxi, which was greatly affected by seasons, and the degree of water stress was relatively serious [36].

## 5. Conclusions

In this paper, by studying the variation trends of NDVI and WUE from 2001 to 2010 and from 2011 to 2020, we discussed the variation characteristics of vegetation WUE in this region under the background of vegetation restoration. This study found that the restoration of vegetation on the Loess Plateau since the project of returning farmland to forest has promoted an increase in vegetation WUE, and the increase in WUE from 2011–2020 was more significant than that in 2001–2010. The area of the Loess Plateau with an upward trend of WUE in 2011–2020 is improved compared to the area in 2001–2010, mainly in parts of Inner Mongolia. We found that with the increase in NDVI, the proportion of the area where WUE increased simultaneously was the largest. The precipitation temperature and WUE in this area showed no significant positive correlation. The correlation indicated that the increase in vegetation was the main reason for the increase in WUE. Therefore, for

the Loess Plateau in arid and semiarid regions, these findings deepen our understanding of the carbon–water coupling mechanism of vegetation, therefore offering crucial suggestions for future ecological restoration management and governance and sustainable development of vegetation on the Loess Plateau.

**Author Contributions:** Conceptualization, R.M.; Data curation, R.M. and D.W.; Formal analysis, R.M. and S.L.; visualization, X.C. and X.Y.; Writing—original draft preparation, R.M. and H.W.; writing—review and editing, H.W. and B.L. All authors have read and agreed to the published version of the manuscript.

**Funding:** This research was funded by National Natural Science Foundation of China: 42141007; Special funds for the transformation of scientific and technological achievements in Inner Mongolia Autonomous Region 2021CG0045; National Natural Science Foundation of China (NSFC. General Projects: (Grant No. 41471430).

**Data Availability Statement:** The GPP and ET datasets are available from Google Earth Engine (GEE) at https://developers.google.com/earth$-$engine/datasets/catalog/CAS_IGSNRR_PML_V2_v017, accessed on 26 June 2022. The NDVI datasets are available from GEE at https://developers.google.com/earth$-$engine/datasets/catalog/MODIS_006_MOD13A1, accessed on 26 June 2022. Meteorological data downloaded from the National Meteorological Science Data Center at http://data.cma.cn/, accessed on 26 June 2022. FLUXNET data downloaded at https://fluxnet.org/data/fluxnet2015-dataset/, accessed on 26 June 2022. The SRTM elevation data were obtained from the Interferometric Synthetic Aperture Radar sensor at http://srtm.csi.cgiar.org/, accessed on 26 June 2022.

**Conflicts of Interest:** The authors declare no conflict of interest.

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
