# Peer review of "Distribution and Driving Force of Water Use Efficiency under Vegetation Restoration on the Loess Plateau"

_remotesensing, doi:10.3390/rs14184513_

Round 1
Reviewer 1 Report
1. The literature review should reflect the need for your paper's research. The innovations of your paper stand out in two ways: 1) long time sequences; 2) reflecting spatial heterogeneity. The preface research review you need to elaborate these two pieces separately and not mix them together. Please clarify the meaning of each of the equation symbols used in 2.4
2. In the article 3.2 description of the appearance of a place name description such as Yulin, it is recommended that the occurrence of the place name be identified in Figure 1 or that figure to facilitate the reader to distinguish the conclusion.
3. The meaning of +- in Table 4 is suggested to be described in the table remarks. Significant data are recommended to be marked in black or highlighted with * and also described in the remarks.
4. The meaning of each of the small and large images in the upper left corner of each of your diagrams needs to be clear.
5. The conclusion should mirror the two innovations in your paper that did the work. Then describe the main conclusion without citing specific data, unless it is an important phenomenal conclusion.
6. lease note that in the introduction section, the authors do not give the full name of some abbreviations when they first mention them, but use the abbreviations directly, which can be confusing to the reader. Please ask the authors to give the full name and abbreviation in time for the first mention of the nouns that require abbreviations, and make sure that only abbreviations are used later in the text. Also, some nouns that appear only once, please do not give abbreviations.
7. Both figures and tables must come immediately after the first time it is mentioned in the text, not far away
8. Eliminate multiple references. After that please check the manuscript thoroughly and eliminate all the lumps in the manuscript. This should be done by character rising each reference individually. This can be done by mentioning 1 or 2 phrases per reference to show how it is different from the others and why it deserves mentioning
9. Consult the journal's reference style for the exact appearance of these elements, and use of punctuation and capitalization.
10. Please insert a space between the numeral and the unit. The only two exceptions are % and °C. Similar modifications should be done throughout the text, including all figures and tables.
11. The descriptions of the pictures in the text are too long, please ask the author to reorganize the language to combine the descriptions to make it easier for readers to understand the meaning of the pictures.
Author Response
Response to reviewer 1
Dear anonymous reviewer,
Thank you very much for your enthusiastic help and valuable comments.
- The literature review should reflect the need for your paper's research. The innovations of your paper stand out in two ways: 1) long time sequences; 2) reflecting spatial heterogeneity. The preface research review you need to elaborate these two pieces separately and not mix them together.
Response/action: Thank you for your scrupulous reminding. We have described the long time series and reflecting spatial heterogeneity separately, about the long time series in lines102-107 and about the spatial heterogeneity in lines108-109.
2.Please clarify the meaning of each of the equation symbols used in 2.4
Response/action: Thank you for your scrupulous reminding. I have added notes on lines 184, 192 and 197 separately
- In the article 3.2 description of the appearance of a place name description such as Yulin, it is recommended that the occurrence of the place name be identified in Figure 1 or that figure to facilitate the reader to distinguish the conclusion.
Response/action: Thank you for your scrupulous reminding. I have modified Figure 1 by adding the province boundaries and names
- The meaning of +- in Table 4 is suggested to be described in the table remarks. Significant data are recommended to be marked in black or highlighted with * and also described in the remarks.
Response/action: Thank you for your scrupulous reminding. Remarks have been added in line 270 to mark important information already in Table 4
- The meaning of each of the small and large images in the upper left corner of each of your diagrams needs to be clear.
Response/action: Thank you for your scrupulous reminding. I added " the upper right corner is the image without excluding significant P>0.05 elements " under Figure 6 and Figure 8 respectively
- The conclusion should mirror the two innovations in your paper that did the work. Then describe the main conclusion without citing specific data, unless it is an important phenomenal conclusion.
Response/action: Thank you for your scrupulous reminding. I've removed the description statements with specific data, as described in lines 388-390
- lease note that in the introduction section, the authors do not give the full name of some abbreviations when they first mention them, but use the abbreviations directly, which can be confusing to the reader. Please ask the authors to give the full name and abbreviation in time for the first mention of the nouns that require abbreviations, and make sure that only abbreviations are used later in the text. Also, some nouns that appear only once, please do not give abbreviations.
Response/action: Thank you for your scrupulous reminding. I have turned the first occurrence of the nouns into and entire and abbreviations, only once the nouns are used in their entirety
- Both figures and tables must come immediately after the first time it is mentioned in the text, not far away
Response/action: Thank you for your scrupulous reminding. I've modified the placement of the description charts to ensure that the charts and tables follow the first mention in the text
9.Eliminate multiple references. After that please check the manuscript thoroughly and eliminate all the lumps in the manuscript. This should be done by character rising each reference individually. This can be done by mentioning 1 or 2 phrases per reference to show how it is different from the others and why it deserves mentioning
Response/action: Thank you for your scrupulous reminding. I have adjusted the number of citations appropriately to ensure that their differences are reflected when citing multiple papers
- Consult the journal's reference style for the exact appearance of these elements, and use of punctuation and capitalization.
Response/action: Thank you for your scrupulous reminding. I have corrected the punctuation and capitalization against the correct format
- Please insert a space between the numeral and the unit. The only two exceptions are % and °C. Similar modifications should be done throughout the text, including all figures and tables.
Response/action: Thank you for your scrupulous reminding. insert a space between the numeral and the unit. The rest of the format is modified according to the standard
- The descriptions of the pictures in the text are too long, please ask the author to reorganize the language to combine the descriptions to make it easier for readers to understand the meaning of the pictures.
Response/action: Thank you for your scrupulous reminding. I have combined and modified the long description text to ensure that it is easy for the reader to read and understand
Thanks again for all of your good ideas and suggestion, we appreciate it very much.

Reviewer 2 Report
- The study focused on the distribution and driving force of water use efficiency under vegetation restoration on the Loess Plateau.
- The introduction section is short it has to be extended to include more information regarding the previous studies carried out on this topic, particularly the use of remote sensing data analysis and WUE for evaluating the ecosystem response to global climate change.
- In Figure 1: What is the source of this map? provide more details.
- Provide a flowchart to show the methodology adopted in this study.
- What are the number of scenes covering the study area for each data type in both time periods 2001-2010 and 2011-2020?
- Provide more details regarding the image processing of the remote sensing data analysis (even on GEE) in this study?
- Did you perform any fieldwork in the study area for validation of the remote sensing data analysis? if yes, then provide more details.
Author Response
Response to reviewer 2
Dear anonymous reviewer,
Thank you very much for your enthusiastic help and valuable comments.
1.The introduction section is short it has to be extended to include more information regarding the previous studies carried out on this topic, particularly the use of remote sensing data analysis and WUE for evaluating the ecosystem response to global climate change.
Response/action: Thank you for your scrupulous reminding. I have added information on WUE research and research on WUE for global climate change scholars in 54-59 line and 81-84 line.
“The characteristics of WUE depend mainly on the strength of the coupling of the GPP and ET components, which are important variables linking the carbon cycle and hy-drological cycle of vegetated ecosystems [4]. The research on the long time series of WUE can help to explore the impact of climate change behind the ecosystem changes at a macro level and reveal the mechanism of global changes on the interaction rela-tionship between water and carbon cycles.
”
“For example, Huang et al. [9] found that WUE increases mainly at high latitudes and decreases at middle and low latitudes under the context of current climate change. tang et al. [10] calculated WUE using MODIS data and showed that the latitudinal distribution of WUE increases from subtropical to middle and high latitudes and then decreases with increasing latitude”
2.In Figure 1: What is the source of this map? provide more details.
Response/action: Thank you for your scrupulous reminding. I have added the description of Figure 1 to lines 159-161 of the article
“DEM data were obtained from the Shuttle Radar Topography Mission (SRTM). The SRTM elevation data were obtained from the Interferometric Synthetic Aperture Radar (InSAR) sensor, ( http://srtm.csi.cgiar.org/, assessed on 20 June 2022)”
3.Provide a flowchart to show the methodology adopted in this study.
Response/action: Thank you for your scrupulous reminding. I have added the flowchart for this article in the appendix
4.What are the number of scenes covering the study area for each data type in both time periods 2001-2010 and 2011-2020?
Response/action: Thank you for your scrupulous reminding. I downloaded the 8 days of PML data from the GEE platform and accumulated it for analysis for each year, which contains 46 remote sensing images per year.
5.Provide more details regarding the image processing of the remote sensing data analysis (even on GEE) in this study?
Response/action: Thank you for your scrupulous reminding. I have added the steps for remote sensing image processing and GEE downloading images, see article lines 148-153
“Firstly, the PML_V2 image 1840 scenes passing over the Loess Plateau from 2001 to 2020 were acquired in the GEE platform using the filtering functions of time and space; then the images were processed by cropping and removing the outliers, and the GPP, ET effects were fused for each year using the cumulative method. The PML-V2 model is based on the stomatal conductivity theory coupled with the total primary productivity process.”
6.Did you perform any fieldwork in the study area for validation of the remote sensing data analysis? if yes, then provide more details.
Response/action: Thank you for your scrupulous reminding. I did not conduct fieldwork, but used FLUXNET observations for the validation of the PML data, and have written say in lines 155-157
“To ensure the accuracy of the model simulation results, this thesis was validated using data from the FLUXNET2015 site (https://fluxnet.org/data/fluxnet2015-dataset/, assessed on 26 June 2022).”
Thanks again for all of your good ideas and suggestion, we appreciate it very much.

Reviewer 3 Report
see attached file

Author Response
Response to reviewer 3
Dear anonymous reviewer,
Thank you very much for your enthusiastic help and valuable comments.
This paper is focused on the temporal evolution of water use efficiency (WUE) in the Chinese Loess Plateau (CLP) from 2001 to 2020.WUE is defined as the ratio of Gross Primary Production(GPP) to evapotranspiration(ET).
In this paper,GPP and ET data were acquired utilizing the Google Earth Engine platformPML_V2v0.1.7 data product. Some explanations should be made about this product : Coupled Evapotranspiration and Gross Primary Product (ET being computed as Penman-Monteith-Leuning evapotranspiration : PML), with 500m spatial resolution and 8 days temporal resolution.
Response/action: Thank you for your scrupulous reminding. I have added the introduction to PML data to lines 151-155 of the article.
“The PML-V2 model is based on the stomatal conductivity theory coupled with the total primary productivity process. The parameters were determined and parameterized on 95 global eddy-related flux stations according to different vegetation types, and then extended globally based on land use data.”
lt should be noticed that GPP and ET estimates can also be obtained from standard MODIS products developed by the Numerical Terra Dynamic Simulation Group of the University of Montana (MOD17A2/A3 and MOD16A2/A3). Five global data sources are available for GPP estimation: MOD17 GPP , revised EC-LUE GPP, OCO-2-based SIF product (GOSIF) GPP , GPP based on near-infrared reflectance of vegetation, and PML-V2 GPP(cf. Du et al.,2022).The choice of data source can influence the results .For example, Zheng et al.(2019) using MODIS standard products obtained a mean value of 1.26±0.28 0.98 g Ckg-1 H2O for WUE of CLP during the period 2000-2014.This is significantly higher than values given in this paper (0.98 g Ckg-1 H2O for2000-2010 and 1.22 0.98 g Ckg-1 H2O for 2011-2020).Other publications give lower values. Zhang et al.(2016) made use of CASA model for estimating NPP and of MOD16 product for ET ; they obtained a value of 0.915 g C/mm-m2 for 2000-2010 in CLP.The good correlation observed between WUE estimates and ground measurements can be an argument supporting the choice of PML-V2 product.
Response/action: Thank you for your scrupulous reminding. I actually used MODIS data in the original experiment, but after verifying with the field data, I found that the accuracy was not as good as the PML data, so I finally used the data used in the current article. In addition, I use FLUXNET observations to validate the PML data in the article, which is explained in lines 155-157 of the article.
“To ensure the accuracy of the model simulation results, this thesis was validated using data from the FLUXNET2015 site (https://fluxnet.org/data/fluxnet2015-dataset/, assessed on 26 June 2022).”
Results confirm the increasing trend in both NDVI and WUE in most situations,suggesting apositive effect of vegetation restoration efforts in CLP since 1999.However, the possible influence of precipitation and temperature factors could be further studied. In this study,no significant correlation was found between WUE and climate factors. Conversely,Zhang et al.(2019)concluded that WUE was positively correlated with temperature and negatively correlated with precipitation. Also the correlations between NPP,ET and WUE with NDVI,,Temperature and Precipitation displayed seasonal variations. The study of these correlations during the active growing season may be helpful to explain spatio-tempral variations of WUE.
Response/action: Thank you for your scrupulous reminding. In the article describing the correlation between WUE and meteorological factors in the Loess Plateau, I used the significant P<0.05 image for the calculation, and I did not use the other significant images, and the images of the correlation between WUE and precipitation and WUE and temperature in the top corner of Figure 8 of the article show that indeed WUE shows a negative correlation with precipitation and WUE shows a positive correlation with temperature. correlation. However, the correlation is not significant because very few pixels pass the significance test.
Other remarks :
- Introduction :
《Zhang et al. [ 13] studied the Loess Plateau's vegetation growth over a lengthy timeseries from1982 to 2014 using MODIS NDVI data 》 : MODIS data are available since 2000, not 1982.
Response/action: Thank you for your scrupulous reminding. You have observed very carefully, and it is true that I made a mistake in writing the article here, the original article used AVHRR and MODIS data for the years 1982-2014. I have corrected it in line 96 of the article.
- Table 2 : some values cited in the text are in disagreement with values displayed in the table ; also DNVI in legend of table ;
- Table 3 : same remark ;
Response/action: Thank you for your scrupulous reminding. I have checked the data and added a new category of NDVI and WUE values in Table 2 Table 3 to ensure that what is mentioned in the article has a corresponding data in the table
- 3.4. The Relationship and Correlation between NDVI and WUE :
In the paragraph below Figure 5, 《DNVI》is mentioned 3 times ;
Response/action: Thank you for your scrupulous reminding. I have corrected the entire article where there are spelling errors
-References list :
The attached reference list does not correspond at all with the numbers cited in the text.
Examples :
Cited in text Reference list
Scartazza et al. [10] 50.Scartazza et al.
Zhao et al. [11] 8. Zhao et al.
Zhang et al.[13] not listed
Xin et al.[14] not listed
Liu et al. [15] 2. Liu et al.
3.Liu et al.
45.Liu et al.
Response/action: Thank you for your scrupulous reminding. I have corrected the full reference to ensure that there are no problems with the citation
Thanks again for all of your good ideas and suggestion, we appreciate it very much.

Round 2
Reviewer 2 Report
Accept in present form